# CT Imaging of Eustachian Tube Balloon Dilation: Method Development on Cadaver Heads

**DOI:** 10.3390/bioengineering10050592

**Published:** 2023-05-14

**Authors:** Selma Cetin-Ferra, Miriam S. Teixeira, J. Douglas Swarts, Tanya J. Rath, Cuneyt M. Alper

**Affiliations:** 1Department of Cell Biology, University of Pittsburgh School of Medicine, Pittsburgh, PA 15260, USA; 2Graduate Medical Education Research Division, Arnot Ogden Medical Center, Elmira, NY 14905, USA; 3Department of Otolaryngology, University of Pittsburgh School of Medicine Pittsburgh, PA 15213, USA; 4Department of Radiology, Division of Neuroradiology, Mayo Clinic College of Medicine, Phoenix, AZ 85054, USA; 5Division of Pediatric Otolaryngology, UPMC Children’s Hospital of Pittsburgh, 1 Children’s Place, 4401 Penn Avenue, Pittsburgh, PA 15224, USA

**Keywords:** eustachian tube, eustachian tube dysfunction, balloon dilation of eustachian tube, imaging, CT imaging, cadaver

## Abstract

*Objective*: To develop a methodology for the measurement of balloon dilation (BD) effects on Eustachian Tube (ET) structure using Computerized Tomography (CT) images. *Methods*: The BD of the ET was performed on three cadaver heads (five ears) through the nasopharyngeal orifice. The axial CT images of the temporal bones were obtained before dilation, while an inflated balloon was in the lumen of ET, and after balloon removal in each ear. Utilizing Dicom images captured by the ImageJ software 3D volume viewer function, the anatomical landmark coordinates of the ET were matched with their pre- and post-dilation counterparts, and the longitudinal axis of the ET was captured with serial images. The histograms of the regions of interest (ROI) and three different lumen width and length measurements were obtained from captured images. The densities of air, tissue, and bone were determined with histograms as a baseline to determine the BD rate as a function of increased air in the lumen. *Results*: The small ROI box included the area of prominently dilated ET lumen after BD and best represented the visually obvious changes in the lumen, compared to the ROIs that extended the wider areas (longest and longer). Air density was the outcome measure for comparison with each corresponding baseline value. The average increase in air density in the small ROI was 64%, while the longest and long ROI boxes showed 44 and 56% increases, respectively. *Conclusion:* This study describes a method to image the ET and quantify the outcomes of BD of the ET using anatomical landmarks.

## 1. Introduction

The Eustachian tube (ET) is an anatomical connector structure between the middle ear (ME) and nasopharynx (NP) [1]. The ET opening is the main means of pressure regulation in the ME and is impaired in patients presenting with otitis media with effusion (OME) [2]. Standard medical therapy for Eustachian tube dysfunction (ETD) is the insertion of ventilation tubes (VTs) to temporarily bypass the ET, equalize ME pressure with the environment, and prevent fluid recurrence. Although widely used, VTs come with potential inconveniences such as otorrhea and rare complications and sequelae such as tympanic membrane perforation, tympanosclerosis, and cholesteatoma, making management of ETD a challenge for otolaryngologists [3]. Balloon dilation (BD) of the ET (BDET) was introduced as an alternative and feasible procedure to treat ETD, with promising outcomes [4,5,6,7]. There is still no consensus for patient selection for BDET or for objective outcome measures to evaluate the long-term effect of BDET [8,9]. Our recent clinical trial in patients with ETD aimed to quantify ET function and structural changes using Computerized Tomography (CT) images that were captured before and after BDET [10]. A pilot study was performed to develop methodology for verification of the anatomical landmarks and the feasibility of quantification of the changes in ET structure utilizing a balloon catheter in the ET lumen for guidance on CT scan X-ray images. The eventual aim was to use this method to evaluate the effect of BDET as a measurable outcome.

## 2. Materials and Methods

This protocol was approved by the Committee for Oversight of Research and Clinical Training Involving Decedents. Three frozen whole male cadaver heads from the University of Pittsburgh School of Medicine Department of Neurobiology and Office for Oversight of Anatomic Specimens were used.

### 2.1. Specimen Preparation

Four days before the experiment, the cadaver heads were thawed, first at room temperature and then in a 4 °C refrigerator. On the day of the experiment, an exam of the mouth, nose, and throat was performed, including 0° and 45° rigid nasal endoscopies (Hopkins II, 2.7 mm, 18 cm, Karl Storz Endoscopy, Storz Xenon 300 light source, Germany). The external ear canals were cleaned under microscopy, and myringotomies were performed bilaterally in all tympanic membranes.

### 2.2. Experimental Protocol

The cadaver head was placed in the supine position, and a plastic ear probe connected to tubing and a three-way valve were placed in the right ear canal to administer contrast medium to the middle ear.

Under the guidance of a 45° rigid endoscope, a 16 mm long balloon catheter (with maximum and minimum diameters of 7 and 3 mm, respectively) was introduced in the left nostril. (Acclarent Balloon Dilation System, Irvine, CA [https://www.jnjmedtech.com/en-US/companies/acclarent, accessed 10 May 2023] . The pharyngeal opening of the ET was identified, and the balloon advanced into the ET lumen. A baseline axial CT scan acquisition was performed by a helical 64 channel multidetector CT scanner (GE LightSpeed VCT; GE Healthcare; Milwaukee, WI, USA). The CT scan protocol parameters were: mAS = 180, kvP = 100, rotation time = 0.6 s, thickness = 0.625 mm, pitch = 0.969, and FOV = 160 mm. The image field of view extended from the tegmen of the temporal bone through the pterygoid processes.

Next, approximately 4–6 mL of 1:3 diluted contrast (Omnipaque [Iohexol] 180 mg/mL, GE Healthcare Inc., Princeton, NJ, USA) was infused through the probe into the right middle ear, and the valve was closed. A balloon was introduced into the left ET lumen and kept uninflated, and a second CT scan acquisition was obtained. A second balloon was then introduced into the right ET lumen, and a third CT scan acquisition of both ears was performed. Both balloons were then inflated to 12 atm for 1 min, and two additional acquisitions were performed: one while both balloons were inflated and another after deflation. Lastly, both balloons were removed, and one final CT scan acquisition was performed.

### 2.3. Image Analysis

First, axial CT images were formatted into the RadiAnt DICOM viewer (version 1.9.14.7431) to be able to be viewed in the ImageJ program (1.51K National Institutes of Health, Bethesda, MD USA). Images with uninflated balloons on the left side were used as the baseline for the images, while the right side had contrast, which was visible in most axial images although not always in the canal. Secondly, sequential images of each series before, during, and after balloon dilation were viewed in the 3D volume viewer plugin of the ImageJ software (Figure 1). The center of the ET longitudinal axis was saved as a single JPEG image after determining X, Y, and Z coordinates and superior-inferior distances by using the image of the inflated balloon as guidance in each ET. This was followed by 12 sequential images from the superior (cranial) and inferior (caudal) directions of the images recaptured and numbered accordingly to include measurements of the whole lumen dimension.

Third, the corresponding centers of the luminal areas in the pre- and post-dilation conditions were re-imaged by creating a plane in the 3D volume viewer using the medial pterygoid plate, hamulus, and sphenoid spine as landmarks. Twenty-four more total sequential images were captured in similar fashion in the 3D volume viewer for each counterpart.

Finally, vertical rectangles of the regions of interest (ROIs) were created in the ImageJ software at the mid-cartilaginous ET lumen level after rotating all the images to bring lumens in the same direction (Figure 2). Initial small ROIs were chosen in the images with inflated balloons with guidance from the contrast at mid-bead level at the two vertical ends. Then the original rectangle was extended toward the NP orifice until the tip of the torus and at the posterior opening to the tip of the balloon by an equal distance in the number of pixels without including the bony structures (longest box), and the last ROI box was shaped by decreasing 5 pixels from each end of the longest box (long box) (Figure 3). Histograms were constructed in the corresponding ROIs of each pre- and post-dilation series in all ET images (Figure 4), followed by baseline histograms from the nasopharyngeal air, surrounding tissue, and the petrous bone (Figure 5) for the determination of the density range of each structure. The mean gray values of each ROI were also measured with ImageJ software in all series images (Figure 6).

Additionally, three different measurements were taken for the length of the lumen with the guidance of the inflated balloon. The longest length of the lumen measurement was from the tip of the torus to the posterior end where the tip of the balloon was located (1); the second length measurement was the distance from the torus to the end of the widest part of the balloon (2); and the last length measurement was the distance from the widest part of the balloon to the tip of the balloon (3) (Figure 7). Finally, pre- and post-dilation widths were measured at 3 different levels of the lumen: at the nasopharyngeal opening (4), mid-cartilaginous (5), and at the posterior end (6) (Figure 8), using the longest ROI as a constant locator of the points in all ears.

## 3. Results

One of the balloon catheter insertions created a false passage, which was determined before the image analysis and excluded that ET from the data. Other catheter insertions were adequate from the NP orifice to the isthmic end.

The baseline histograms of air, tissue, and bony structures in the representative mid-section images of each ear illustrated the number of pixels distributed on the images for each level (gray value), from the darkest (0) to the brightest (256), for all ROIs that were saved with the ImageJ software, for each described step of imaging. The pixel values in the air density found 0–16 with the baseline histograms that were taken with locating small ROIs in the nasopharyngeal space and areas out of the frame adjacent to the cadaver head. We also checked the pixel values in the dilated portion of the ET lumen in several sequential images. These measurements increased the air density distribution per pixel up to 31 units in the histograms, suggesting there is a partial volume effect in the lumen. Tissue and bone pixel values were also documented to be within the ranges of 42–88 and 214–255, respectively. We have used two different ranges of pixel values for the air density (0–16 and 0–31) in two different calculations as the total number of pixels in all ROIs in the sequential images of each pre- and post-dilation condition in five ears. Small ROI box measurements depicted the change better than the longest and long box ROIs as pixel distribution in the air density, which were observed with clearly widened lumens in post-dilation images compared to the pre-dilation counterparts (Figure 8). The number of pixels in the air density range pre- and post-dilation for all five ears is demonstrated in Figure 9. This representation with small ROIs was expected since the longest and long boxes included more heterogeneous structures. There was a 20–100% increase in air pixel values with small ROI histograms (average 64%), while the longest and long ROI boxes showed an increase between 6–68 and 8–97% (averages 44 and 56%), respectively (Table 1).

ET lumen width measurements also showed increases compared to baseline values, especially the width at the center of the balloon (width 2). The width percentage increase was 57.8–93.5% (width 2, average 71.8%, and SD = 16.4), while NP opening and the lumen at the isthmic end dilated between 0 and 25 (width 1, average 8.3, and SD = 9.9) and 0–100 (width 3, average 37.8, and SD = 39.4%, respectively) (Table 2).

The average of the longest lumen in the five ears was 75.42 (SD = 7.4) pixels. The average lengths of the cartilaginous part and the bony part of the ET lumen were 45.26 (SD = 8) and 31.26 (SD = 6.5) pixels, respectively, which are approximated with the distance 2 and 3 measurements as an average in the five ears (Table 3).

## 4. Discussion

Balloon Dilation Eustachian Tuboplasty (BDET) is being increasingly used for ETD in adults and now children, first in Europe and then in the US, with the intention of decreasing and/or eliminating the need for long-term ventilation tubes in chronic conditions such as chronic OME [6,7,11,12,13,14]. Evaluations of BDET effects are mostly subjective, involving clinical outcome scoring before and after the procedure using a questionnaire such as the Eustachian Tube Dysfunction Questionnaire-7 (ETDQ-7) and/or the Eustachian Tube Score (ETS), in some studies combined with Valsalva and Toynbee maneuvers [15,16]. Tympanometry has been widely used as part of the diagnostic tools for ETD for comparison before and after the procedure [5,6,7]. The R value measured with tubomanometry while swallowing water (R ≤ 1 immediate opening, R >1 late opening, non-measurable R closed tube) was recommended as a functional outcome measure for BDET, indicating latency of the ET [17]. In our previous study, we found that the presence of an R value had high sensitivity for detecting an ET opening but low specificity for detecting a non-opening [18]. For this reason, the tubomanometry test by itself may not be an ideal objective measure of the BDET outcome.

The ETDQ-7 is widely used by clinicians, and a recent randomized study indicated that BDET is significantly superior to medical management for persistent ETD [8], but our previous study showed it is only moderately associated with an objective measure of ET function and appears less reliable when applied to people with non-intact tympanic membranes [19]. We think scoring systems should be supported by functional and structural objective measures for the evaluation of BDET effects to confirm its long-term benefits in clinical trials.

Visualization of the ET in imaging studies has always been a challenge due to its normally closed status and small size, unless it is patulous [20]. The first CT studies in a cadaver head and/or live images were mostly performed to determine the length, volume, and angle of the ET lumen for better understanding ET structure and function [10,21,22,23,24]. Nasopharyngeal maneuvers such as Valsalva were added to the scans to get a clear opening, but capturing the ET was still complicated by the lack of contrast in the tissues. Three-dimensional (3D) CT imaging studies were very beneficial for the understanding of temporal bone structure and for the diagnosis and follow-up of malignant tumors [25,26,27,28,29].

CT imaging of the ET has been part of the pre-procedure evaluation for BDET to determine if there is dehiscence and abnormality of the carotid artery and to prevent potential lethal complications [23,29,30] CT imaging preoperatively has been part of the protocol for BDET for most surgeons, and sometimes it is obtained after the procedure to see if there is any damage to the structures. A study showed no predictivity of difficulties and complications with preoperative high-resolution CT scanning, and these same authors suggested that the fear of injury to the internal carotid artery during the procedure might be disproportionate since there has not been published data about this complication [23]. Because physicians feel safer visualizing ET structure preoperatively, adding a quantification technique to the image analysis that helps to see the correlation between dilated lumen and symptoms could magnify the benefit of CT exposure pre- and post-operatively. We can also synchronize these imaging studies with the forced response test (FRT), as we established the method in our previous pilot study with five non-intact ears and/or some of the nasopharyngeal maneuvers to capture cartilaginous lumen opening before and after BDET [10]. Measurements would be performed in Hounsfield Units (HU) as in the later study, or histograms can be counted as pixel values after creating the area of interest as detailed in the methods in this paper. The idea is to compare air and tissue intensities in pre- and post-procedure images after determining the baseline values of the known anatomical areas, such as the nasopharyngeal space for air, surrounding tissues, and petrous bone intensities. However, HU is a universal, appropriately standardized metric for the analysis of CT images across scanner models. Even though in the current study, conversion of HU to the non-standard ranges (0, 255) for tissue types has been utilized with ImageJ software, utilizing a 16 or 32 bit image may allow preservation of the previously accepted ranges for various tissue types.

In this cadaver study, we have used contrast instilled through a non-intact tympanic membrane in an attempt to picture the ET before catheter insertion. Although contrast was visible in most images, the time it took to pass the contrast was too short to be able to visualize the entire length of the ET lumen, even with synchronized imaging in the cadaver heads. Using CT images with the inflated balloon served the purposes of determining feasibility and finding anatomical landmarks while viewing the whole anatomy of the head in 3D without a need for contrast locally or systemically. Even though there was some degree of potential mismatch between the pre- and post-dilation CT images, when there were all three landmarks in one plane, it was possible to approximate the counterparts to capture reasonably comparable images at the same plane of anatomy.

While many case series have suggested BDET can successfully restore ET function, systematic reviews have indicated no high-quality evidence exists to support its effectiveness [8,9,31]. Recently, otolaryngologists have been focused on creating a consensus for the eligibility criteria of ETD for BDET and the objective evaluation of its effects with more randomized clinical trials [32].

One study showed CT scanning in the sitting position may visualize ET opening better, especially in patients with a patulous ET [33]. These authors also suggested that vertical CT, which is mainly used in dental clinics, can be used to visualize the ET at a low cost and with less radiation exposure than conventional temporal bone CT imaging [29]. Since we are proposing an additional postoperative CT for BDET, it would be important to decrease the cost and radiation exposure of vertical CT imaging and increase its effectiveness by accommodating the position while scanning.

The proposed image analysis and quantification method with ImageJ is relatively practical for the clinician, and the software is freely available from the NIH. This was our choice of image analysis software due to its availability and our familiarity with it from our previous work. However, there is other, more advanced image analysis software available that may be more user-friendly. Some options may be able to generate ROIs, and most will be able to get various summary data from these ROIs. Even though explicit histograms may or may not be possible with a specific software program, such a feature may not be needed with some study material. Although a time-consuming learning curve is the downside of the ImageJ method, there is the advantage of more than one obtainable measure in matched pre- and post-procedure images of the ET. Additionally, we could depict the increase in width measurements with visually prominent lumen dilation in all ears in the CT images. The length measurements were also comparable to those in the previous literature; the cartilaginous part and the bony part of the ET lumen averaged 45.26 (16.8 mm) and 31.26 (11.3 mm) pixels, respectively, displaying the precision of the technique [28]. The mean gray value was one of the useful measures that can be performed in ImageJ software, and representation as a graph is also available for future studies for the confirmation of dilation rates. We previously presented the ETF test data from the clinical BDET study with eleven adult subjects [14] and are in the process of evaluating the pre- and post-procedure CT images in the same data set with the proposed method. Of course, we did not have the advantage of having the balloon in the canal during inflation on these live images, and dilations were not as prominent as in the cadaver ETs without tissue elasticity, but we found this image analysis method applicable, having three landmarks (medial pterygoid plate where the tissue thickness is the minimum, immediate opposite side medial pterygoid plate hamulus, and sphenoid spine superiorly) in the same plane by the 3D viewing tool. This could be an important part of the quantification of the effect of BDET in clinical research studies and aid physicians in future patient selection and evaluation of BDET outcomes. Future studies should also take into account tissue elasticity, recoil, and elastic fiber remodeling during and after the application of such pressure on the ET with various tissue properties including cartilage, muscle, fat, and possibly inflamed and thickened mucosa. Moreover, even though ET dilation balloons do reach a maximum diameter of 6 mm at pressures of 10 ATM, the biomechanics of the balloons would need to be considered.

## 5. Study Limitations

The main limitation of this study is the small sample size. Even for method development, a larger sample size would have been better;The ImageJ image analysis software may have limitations compared to other newer software that may eliminate some of the labor-intensive steps in standardizing ROIs and generating outcome images and reports;Utilizing a cadaver head in the study has inherent limitations due to tissue elasticity and recoil properties similar to those in live human subjects. On the other hand, it was possible to have six consecutive CT scans as per the protocol on cadaver heads for the purpose of method development, while this would not be justified in live human subjects;The effects of balloon shape and length, as well as different durations of balloon inflation, have not been explored in this study. Future studies incorporating and controlling the biomechanics of the balloons and their effects on tissues may provide a better understanding of the role of this treatment and its long-term outcomes and consequences.

## 6. Conclusions

Imaging of the ET, particularly comparison of the effect of an intervention, has inherent challenges due to the technique and image analysis. This study is an attempt to describe a method to image the ET and quantify the outcomes of BD of the ET using anatomical landmarks.

## Figures and Tables

**Figure 1 bioengineering-10-00592-f001:**
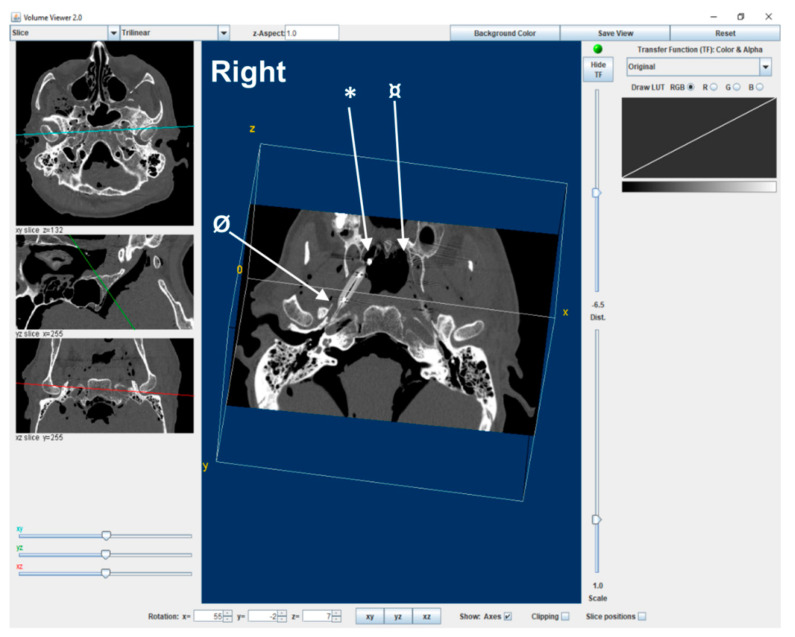
Example of 3D volume viewing of a CT image in ImageJ software under the guidance of anatomical landmarks *(** *medial pterygoid plate spine*, *¤ opposite medial pterygoid plate spine*, and *Ø sphenoid spine)* and an inflated balloon to determine the ET longitudinal axis coordinates at the center as a starting point of image analysis. The X, Y, and Z coordinates of the image on the right are demonstrated on the smaller images on the left side as blue line (top), green line (middle image), and red line (bottom image).

**Figure 2 bioengineering-10-00592-f002:**
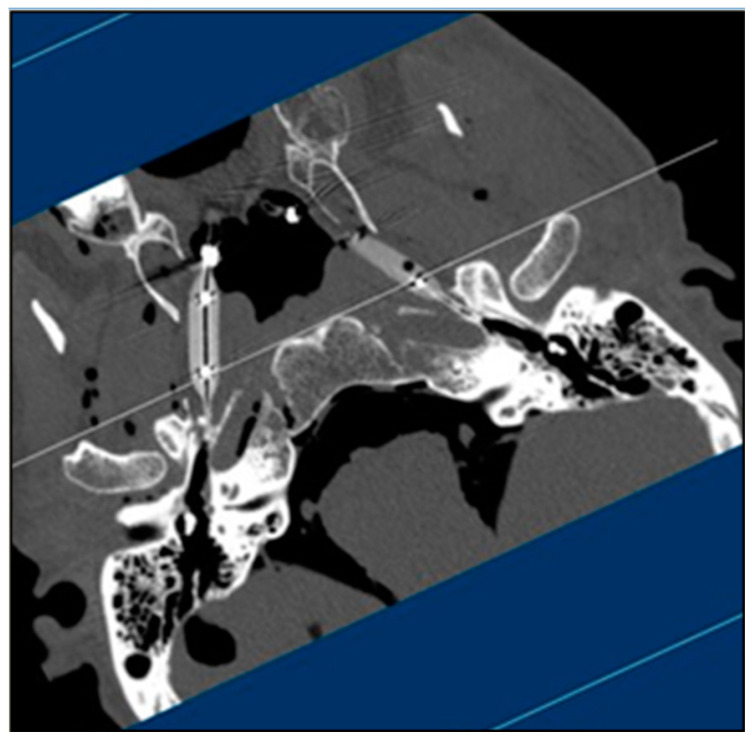
Example of a vertically rotated JPEG image captured at the center of the longitudinal axis from the NP to the isthmus after positioning in ImageJ software (Ear 1—right).

**Figure 3 bioengineering-10-00592-f003:**
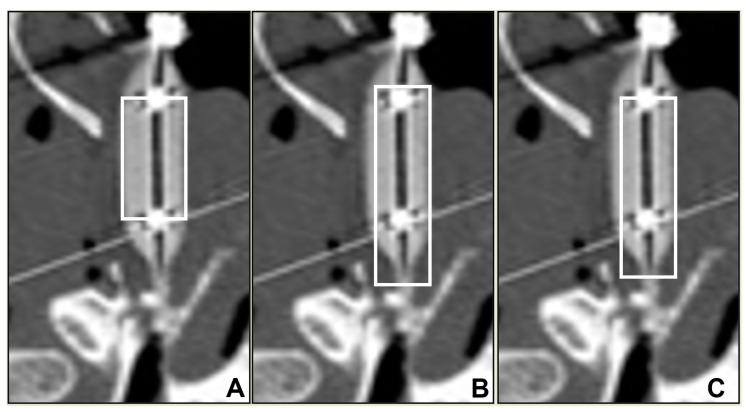
Example of how vertical ROI boxes (white rectangle) (**A**)—small; (**B**)—longest; (**C**)—long) are placed in an inflated balloon-guided ET lumen (Ear 1). Outline of the small ROI boxes (A) was formed to include the main body of the inflated balloons, to include the 6 mm width in the full 16 mm cylindrical segment. Radiopaque markers were used as markers, and short sides of the ROI boxes were set to cross these markers. This segment of the balloon represents the section of the ET that was subjected to the maximum dilation pressure. Proximal and distal ends of the balloons had conical sections which had a relatively less dilation force on the tissues. In order to include a longer section of the ET in the ROI, and to differentiate the effect of these cylindrical and conical sections of the balloons on the tissues, the longest ROI box (B) was extended to the end of the balloon towards the middle ear (bottom of the images) and towards the nasopharyngeal end of the balloon (top of the images) while avoiding inclusion of nasopharyngeal air column. The long ROI (C) was then formed by retracting the box by 5 pixel units from the middle ear and the NP end.

**Figure 4 bioengineering-10-00592-f004:**
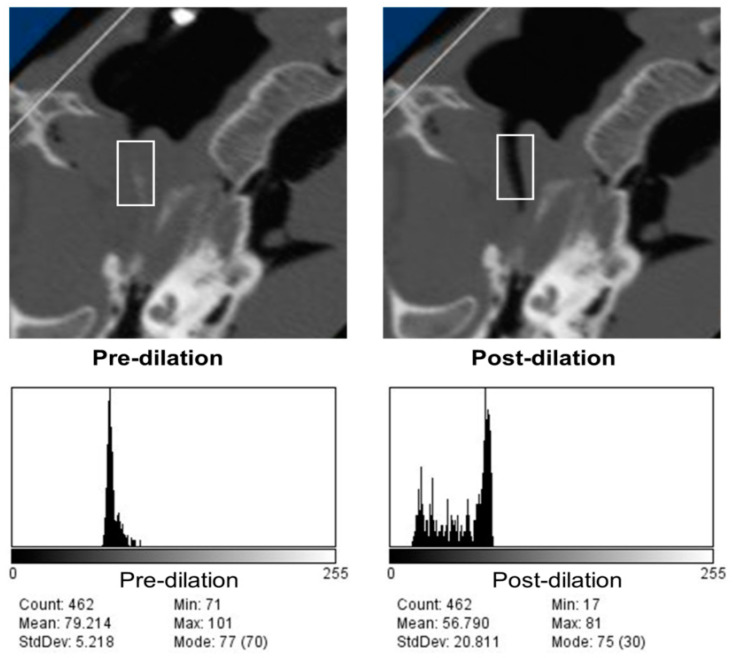
Small ROI placements (white rectangle boxes) in the center of the longitudinal axis of the lumen (**top row**); histograms of the ROIs shown in the top row (**bottom row**).

**Figure 5 bioengineering-10-00592-f005:**
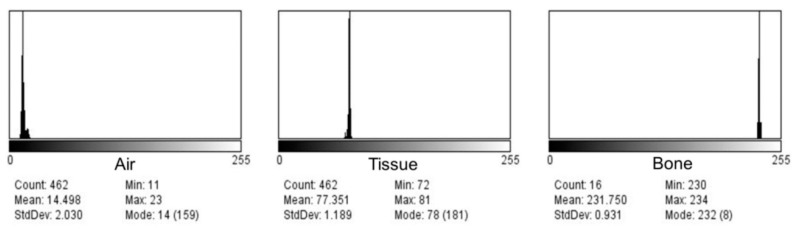
Baseline histograms in nasopharyngeal air, tissue, and bone (Ear 4).

**Figure 6 bioengineering-10-00592-f006:**
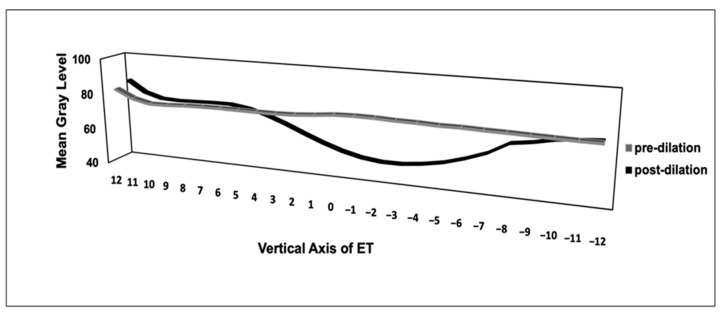
Representative graph for the measurement of mean gray values in small ROIs in pre- and post-dilation images in ET (Ear 4). The graph with mean gray value depicts markedly the visually observed dilation in the serial CT images of the ears pre- versus post dilation image of this example ear.

**Figure 7 bioengineering-10-00592-f007:**
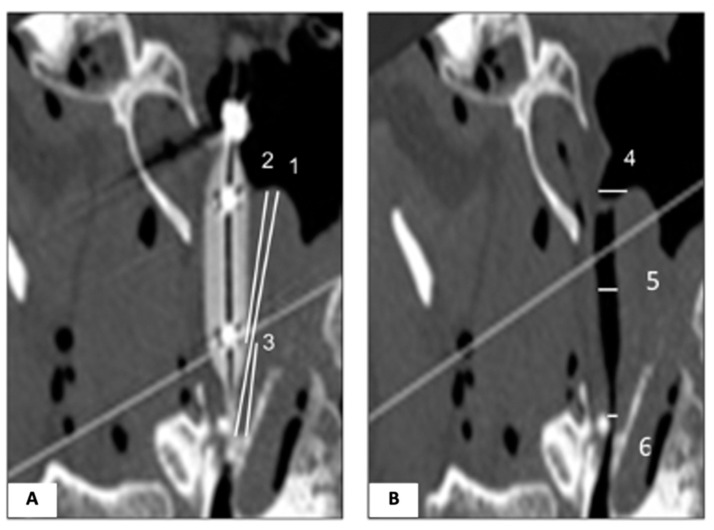
Measurements of ET structure under the guidance of the balloon and ROI. (**A**) Image during dilation at the center of the balloon lumen, representing the center of the longitudinal axis of the ET (Ear 1). The longest luminal length (distance 1), the length from the torus to the end of the widest part of the balloon at the tip (distance 2), and the length from the widest part of the balloon to the tip of the balloon (distance 3). (**B**) Image after dilation at the center of the longitudinal axis of the ET (Ear 1). NP opening at the torus #4 (width 1), the width at the mid-balloon level #5 (width 2), and the width at the isthmus end of the Eustachian lumen #6 (width 3).

**Figure 8 bioengineering-10-00592-f008:**
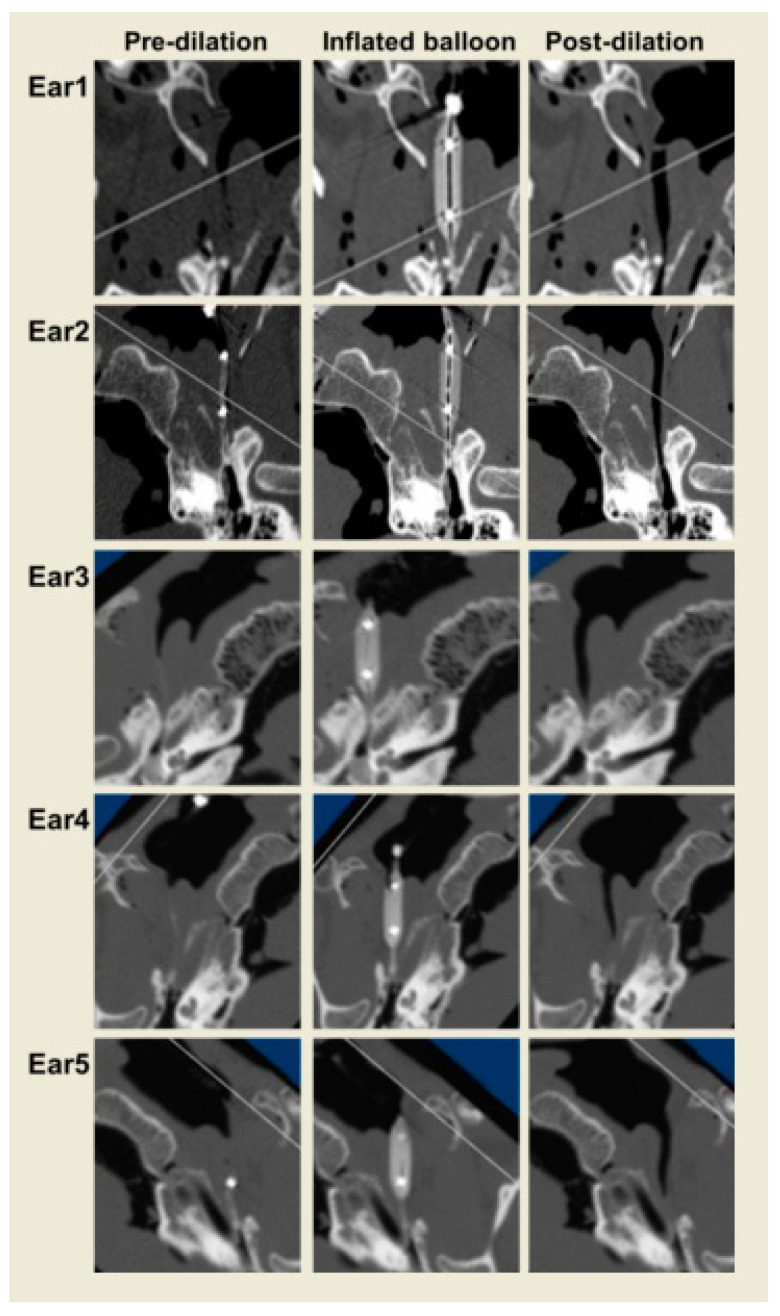
Center of the ET longitudinal axis in images pre-, during, and post-dilation in each ear (Ear 1–Ear 5).

**Figure 9 bioengineering-10-00592-f009:**
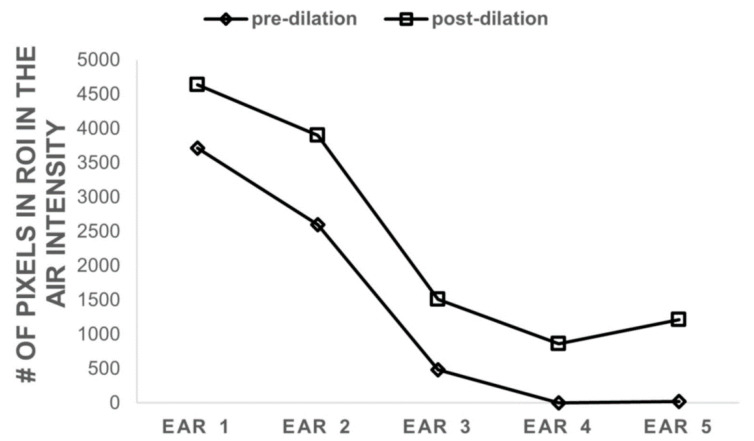
Total pixel values in the ROI (small box) in the air density values (0–31) pre- versus post-dilation in each ear.

**Table 1 bioengineering-10-00592-t001:** Percentage increase in air density (pixel unit) pre- versus post-dilation in all three ROIs in each ear. There was a 20–100% increase in air pixel values with small ROI histograms (average 64%), while the longest and long ROI boxes showed an increase between 6–68 and 8–97% (averages 44 and 56%), respectively.

	Small	Longest	Long
Ear 1	20	6	8
Ear 2	33	23	26
Ear 3	68	50	62
Ear 4	100	74	97
Ear 5	98	68	88
% Increase	64	44	56

**Table 2 bioengineering-10-00592-t002:** Percentage increase in lumen width (pre- versus post-dilation) in all ears.

	Width 1	Width 2	Width 3	Average
Ear 1	3.5	57.8	31.6	31
Ear 2	0	93.5	10	34.5
Ear 3	9	75.6	100	61.5
Ear 4	4.2	53.3	47.2	34.9
Ear 5	25	78.9	0	34.6
Average pixel	8.3	71.8	37.8	39.3

**Table 3 bioengineering-10-00592-t003:** Distance measurements as pixels in all ETs.

	Distance 1	Distance 2	Distance 3
Ear 1	67.99	50.45	20.82
Ear 2	87.74	56.01	34.41
Ear 3	73.37	44.24	30.16
Ear 4	75.41	37.79	38.09
Ear 5	72.6	37.79	32.82
Average (pixel)	75.42	45.26	31.26

## Data Availability

Not applicable.

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
