# Peer review of "CT Imaging of Eustachian Tube Balloon Dilation: Method Development on Cadaver Heads"

_bioengineering, 2023, doi:10.3390/bioengineering10050592_

Round 1
Reviewer 1 Report
The manuscript "CT Imaging of the Eustachian Tube Balloon Dilation: Method Development on Cadaver Heads" is a proof-of-concept study to evaluate the ability for CT imaging to detect changes in eustachian tube patency after balloon dilation.
The aims of the study are clear, and the overall design is appropriate for those aims. A few issues would make the manuscript significantly stronger:
- The "Experimental protocol" section under "Materials and Methods" is difficult to follow. Specifically, the laterality of each step is often not mentioned (e.g. lines 77-83 seem to be all referring to the LEFT side, though this is not explicitly stated)
- The choices involving the image-processing pathway are both complicated and needlessly lossy with respect to data interpretation:
- First, all modern PACS viewers have the ability to generate ROIs (including RadiAnt), and most will be able to get various summary data, from these ROIs. Explicit histograms may or may not be possible, though it is less clear if this will be necessary in a final pipeline. If the goal is a clinical processing pathway, the additional exporting/reformatting adds unnecessary complexity
- Second, Hounsfield Units (HU) is a universal, appropriately standardized, metric for analysis of CT images across scanner models. The conversion to the range [0, 255] is lossy and requires non-standard ranges for tissue types (lines 172-177). Even if ImageJ is the desired analysis program, utilizing a 16- or 32-bit image would allow preservation of the previously-accepted ranges for various tissue types. This is even briefly suggested in line 242
In summary, this pilot study is very interesting as it teases a clinical imaging pathway to evaluate one metric of success in estuation tube balloon dilation. A few clarifications and small changes to the processing would significantly improve the manuscript applicability.
Author Response
Dear Editor,
We appreciate the comments and suggestions of the Reviewer 1. We have addressed the comments (in bold and italics) point by point with our responses below.
REVIEWER 1
The manuscript "CT Imaging of the Eustachian Tube Balloon Dilation: Method Development on Cadaver Heads" is a proof-of-concept study to evaluate the ability for CT imaging to detect changes in eustachian tube patency after balloon dilation.
The aims of the study are clear, and the overall design is appropriate for those aims. A few issues would make the manuscript significantly stronger:
Authors appreciate the positive comments of the Reviewer.
- The "Experimental protocol" section under "Materials and Methods" is difficult to follow. Specifically, the laterality of each step is often not mentioned (e.g. lines 77-83 seem to be all referring to the LEFT side, though this is not explicitly stated)
Authors appreciate the Reviewer’s attention to our lack of clarity in the steps of the Experimental Protocol. We did edit the section and added information for more clarity.
- The choices involving the image-processing pathway are both complicated and needlessly lossy with respect to data interpretation:
Authors appreciate this comment. We have chosen to have a detailed description of the image processing pathway on order to facilitate reproducibility, if needed, and also for applicability of the methods on other imaging material.
- First, all modern PACS viewers have the ability to generate ROIs (including RadiAnt), and most will be able to get various summary data, from these ROIs. Explicit histograms may or may not be possible, though it is less clear if this will be necessary in a final pipeline. If the goal is a clinical processing pathway, the additional exporting/reformatting adds unnecessary complexity
Authors appreciate the comment. Image J was the image analysis software that we had in our laboratory. We do not have access to the newer software, and unfortunately, we do not have funding to support the acquisition of any of those. However, we did include in the discussion section, the availability of other software with the advantages stated by the Reviewer.
- Second, Hounsfield Units (HU) is a universal, appropriately standardized, metric for analysis of CT images across scanner models. The conversion to the range [0, 255] is lossy and requires non-standard ranges for tissue types (lines 172-177). Even if ImageJ is the desired analysis program, utilizing a 16- or 32-bit image would allow preservation of the previously-accepted ranges for various tissue types. This is even briefly suggested in line 242
Authors appreciate the Reviewer’s comments. We have added the suggested information into the relevant section in the Discussion.
In summary, this pilot study is very interesting as it teases a clinical imaging pathway to evaluate one metric of success in estuation tube balloon dilation. A few clarifications and small changes to the processing would significantly improve the manuscript applicability.
Authors appreciate the positive comments and thankful for the revision suggestions that we believe have improved our manuscript.
Reviewer 2 Report
In this manuscript authors described a method to image the ET and to quantify the outcomes of BD of the ET using anatomical landmarks. There is no limitation of the study. Why only anatomical measurement are performed? Why elastic component of the tissue was not take into account? Also biomechanics of the balloon was not considered.
Author Response
Dear Editor,
We appreciate the comments and suggestions of the Reviewer 2. We have addressed the comments (in bold and italics) point by point with our responses below.
REVIEWER 2
In this manuscript authors described a method to image the ET and to quantify the outcomes of BD of the ET using anatomical landmarks. There is no limitation of the study.
Authors appreciate the comment.
Why only anatomical measurement are performed? Why elastic component of the tissue was not take into account?
Authors appreciate the comment. We know cadaver head is lacking normal elasticity but the purpose of the study was creating an applicable method to live tissue with determined anatomical landmarks. We had emphasized the weakness of this method using cadaver heads in lines 282-284 as “Of course, we did not have the advantage of having the balloon in the canal during inflation on these live images and dilations were not as prominent as in the cadaver ETs without tissue elasticity,…”.
Also biomechanics of the balloon was not considered.
Authors appreciate the comment. We have added a sentence indicating the need for considering biomechanics of the balloons used for the ET dilation.
Round 2
Reviewer 1 Report
The authors have addressed my concerns and I recommend publication.
Author Response
Authors appreciate the reviewer's favorable review. Spell-check has been performed.
Reviewer 2 Report
Still limitation of the study are not mentioned.
Author Response
Authors appreciate the comments of Reviewer 2. We have included a section that lists the study limitations. We also had the manuscript reviewed and edited by a native English speaker.